# Molecular Epidemiology of *Escherichia coli* Clinical Isolates from Central Panama

**DOI:** 10.3390/antibiotics10080899

**Published:** 2021-07-23

**Authors:** Virginia Núñez-Samudio, Maydelin Pecchio, Gumercindo Pimentel-Peralta, Yohana Quintero, Mellissa Herrera, Iván Landires

**Affiliations:** 1Instituto de Ciencias Médicas, Las Tablas 0701, Los Santos, Panama; vnunez@institutodecienciasmedicas.org (V.N.-S.); angel_mpm@yahoo.com (M.P.); gumercindo1441@gmail.com (G.P.-P.); jhnquintero53@gmail.com (Y.Q.); 2Sección de Epidemiología, Departamento de Salud Pública, Región de Salud de Herrera, Ministry of Health, Chitré 0601, Herrera, Panama; 3Unidad de Infectología, Hospital Dr. Gustavo Nelson Collado, Caja de Seguro Social, Chitré 0601, Herrera, Panama; 4Laboratorio Clínico, Hospital Luis “Chicho” Fábrega, Región de Salud de Veraguas, Ministry of Health, Santiago 0923, Veraguas, Panama; myherrera@minsa.gob.pa; 5Centro Regional Universitario de Azuero (CRUA), Universidad de Panamá, Chitré 0601, Herrera, Panama; 6Hospital Joaquín Pablo Franco Sayas, Región de Salud de Los Santos, Ministry of Health, Las Tablas 0701, Los Santos, Panama

**Keywords:** molecular epidemiology, *Escherichia coli*, Panama, pandemic clone ST43/ST131, CTX-M-15

## Abstract

*Escherichia coli* represents one of the most common causes of community-onset and nosocomial infections. Strains carrying extended spectrum β-lactamases (ESBL) are a serious public health problem. In Central America we have not found studies reporting the molecular epidemiology of *E. coli* strains implicated in local infections, so we conducted this study to fill that gap. *Materials and Methods:* We report on an epidemiological study in two reference hospitals from central Panama, identifying the susceptibility profile, associated risk factors, and molecular typing of *E. coli* strains isolated between November 2018 and November 2019 using Pasteur’s Multilocus Sequence Typing (MLST) scheme. *Results:* A total of 30 *E. coli* isolates with antimicrobial resistance were analyzed, 70% of which came from inpatients and 30% from outpatients (*p* < 0.001). Two-thirds of the samples came from urine cultures. Forty-three percent of the strains were ESBL producers and 77% were resistant to ciprofloxacin. We identified 10 different sequence types (STs) with 30% of the ESBL strains identified as ST43, which corresponds to ST131 of the Achtman MLST scheme—the *E. coli* pandemic clone. Thirty-eight percent of the *E. coli* strains with the ESBL phenotype carried CTX-M-15. *Conclusions:* To the best of our knowledge, this is the first report confirming the presence of the pandemic *E. coli* clone ST43/ST131 harboring CTX-M-15 in Central American inpatients and outpatients. This *E. coli* strain is an important antimicrobial-resistant organism of public health concern, with potential challenges to treat infections in Panama and, perhaps, the rest of Central America.

## 1. Introduction

*Escherichia coli* represents one of the most frequent causes of bacterial infections [1] and it accounts for 70% to 95% of community-onset acute urinary tract infections (UTIs) and 50% of nosocomial infections [2]. β-Lactam antibiotics and fluoroquinolones are widely prescribed to treat both community- and hospital-based infections caused by *E. coli* [3]. However, resistance to these categories of antibiotics has increased worldwide, which represents a major public health problem. For example, it has been reported that third-generation cephalosporins and fluoroquinolones have registered resistance rates greater than 50% in five of the six working regions of the World Health Organization (WHO) [1,4,5].

Among the *E. coli* strains, the principal mechanism of resistance to β-Lactams is the production of β-lactamase enzymes, all of which differ from each other based on their substrate profile, inhibitor profile, and sequence homology [5]. Extended-spectrum β-lactamases (ESBLs) are a group of enzymes that cause resistance to oxyminocephalosporins (i.e., cefotaxime, ceftazidime, cefuroxime, and cefepime) and monobactams (i.e., aztreonam), but not to cefamycins (i.e., cefoxitin) or carbapenems (i.e., imipenem, meropenem, and ertapenem) [5]. Several risk factors associated with ESBL-producing *E. coli* infections have been described in the community, including: previous use of antibiotics (especially quinolones and third- or fourth-generation cephalosporins), recurrent *E. coli* infections, recent hospitalization (within the prior year), artificial nutrition, previous admissions to intensive care units (ICUs), foster home stays, and receiving hemodialysis [6,7,8]. Strains of ESBL-producing *E. coli* are important antimicrobial-resistant organisms (AROs). In 2017, the WHO defined a list of priority AROs for research purposes, among which ESBL-producing *Enterobacteriaceae* are in priority group 1 [9].

The *E. coli* sequence type 43 (ST43) in Pasteur’s multilocus sequence typing (MLST) scheme—which corresponds to ST131 in the Achtman MLST scheme—constitutes a pandemic clonal lineage, responsible for a large number of multidrug resistant infections worldwide. The strains belonging to the ST131 clone differ in serotypes O25b and O16, which correspond, respectively, to STs PST43 and PST506 when using Pasteur’s MLST scheme [10,11]. Global emergence of *E. coli* ST131 isolates occurred approximately 30 years ago, most likely in a North American context and consistent with strong selection pressure exerted by the widespread introduction and use of extended-spectrum cephalosporins and fluoroquinolones [12,13]. The member strains of ST 131O25b (PST43) are disseminated worldwide and commonly reported in extraintestinal infections as producers of the ESBLs enzymes known as CTX-M-15 or CTX-M-14, which also exhibit resistance to fluoroquinolones [14,15,16,17]. On the basis of their amino acid sequence similarities, the CTX-M family have been classified into five major groups, named CTX-M group-1, -2, -8, -9, and 25/26. CTX-M-15 (of the CTX-M-1 group) and CTX-M-14 (of the CTX-M-9 group) are the most frequent enzymes isolated. The CTX-M-15 β-lactamase is the dominant ESBL in ST131 and is increasingly found in isolates causing both UTI and bacteremia [14]. This dissemination has increased both in the hospital environment and in the community [18]. Some studies from Latin American countries have described the presence of ST131, both in hospitals and the community [19,20,21,22]. In Ecuador, a study of *E. coli* serotype O25 carrying ESBL belonged to ST 131/PST43 and resulted in carriers of CTX-M-15 [23]. However, to the best of our knowledge, there are still no reports on the molecular epidemiology of infection-causing *E. coli* strains in Central America [24]. The purpose of this study was to characterize strains of *E. coli* implicated in infections in hospitalized and outpatients in two reference hospitals from central Panama. By analyzing the *E. coli* isolates from hospital-based clinical laboratories we aim to identify the susceptibility profile, associated risk factors, characterization of ESBL-producing strains and molecular typing using Pasteur’s MLST scheme.

## 2. Materials and Methods

### 2.1. Study Setting

We conducted a prospective epidemiological study between November 2018 and November 2019 in two reference hospitals in central Panama: The Cecilio Castillero General Hospital (CCGH) and the Luis “Chicho” Fábrega Hospital (LCFH), located in the provinces of Herrera and Veraguas, respectively. Both hospitals are public assistance hospitals and represent the main centers that provide medical and laboratory care in the central region of Panama.

### 2.2. Isolates of E. coli

During the study period, we included *E. coli* samples that (a) were isolated from diverse samples from outpatients and hospitalized patients, within the first 48 h of admission, (b) were collected as part of routine patient care procedures, and (c) showed resistance to at least one of the antibiotics routinely tested in the hospitals’ clinical microbiology laboratories. The in vitro antimicrobial activity was determined using the Vitek 2 system through Gram-negative susceptibility cards AST-GN69 (Ref. 413 400) and AST-N250 (Ref. 413 573) (BioMérieux; Marcy l’Etoile, France). The test results were interpreted according to breakpoints defined by the Clinical and Laboratory Standards Institute (CLSI) [25].

A technical sheet was completed anonymously for each sample collected, recording the following risk factors: age, sex, hospitalization for 2 or more days in the prior 90 days, antibiotic treatment in the prior 90 days, personal history of immunosuppressive therapy (International Classification of Diseases ICD-10, code Z92.25), wound care at home, hemodialysis within the prior 90 days, and outpatient chemotherapy. The data recorded were captured in MS Excel (The Microsoft Corporation; Redmond, WA, USA). Data analyses were conducted in Stata v. 11.0 (StataCorp, LLC; College Station, TX, USA). We calculated descriptive statistics and estimates with their respective 95% confidence intervals (Cis). We used Fisher’s exact test to compare proportions, setting alpha at 0.05 for statistical significance when comparing the frequencies of ESBL- and non-ESBL-producing isolates.

### 2.3. Molecular Typing Analyses and β-Lactamase Molecular Identification

Molecular typing analyses were performed using Pasteur’s MLST scheme. We conducted MLST schemes using a standardized protocol specific for *E. coli* [26]. The sequencing internal fragments of eight housekeeping genes (i.e., *dinb*, *icdA*, *pabB*, *polB*, *putB*, *trpA*, *trpB*, and *uidA*) were amplified from chromosomal DNA of the *E. coli* strain. Sequencing of the polymerase chain reaction (PCR) products was performed using the services of Macrogen (Macrogen Inc.; Seoul, Korea). Sequences of the genes were analyzed by Geneious prime v. 2020.5 (Biomatters, Ltd.; Auckland, New Zealand) and the allelic profile was determined using *E. coli* MLST datababes (https://bigsdb.pasteur.fr/ecoli/ecoli.html, accessed on 03 May 2021) [27].

All isolates with phenotype ESBL were screened for *bla*SHV, *bla*TEM and *bla*CTX-M gene families by a PCR assay using specific primers as previously described [28]. *bla*CTX-M groups 1, 2, 8, 9, 25 and group 1 variant (CTX-M-15) were identified by a PCR assay using specific primers as previously described [29,30].

## 3. Results

A total of 30 isolates of *E. coli* with antimicrobial resistance were analyzed in this study, 20 (67%) of which came from urine cultures, and 5 each (16%) from blood and wound cultures (*p* < 0.001). Twenty-one samples came from hospitalized patients and nine came from outpatients (*p* < 0.001). In general, most (19) patients were female (*p* < 0.001). The mean age was 57.26 (95% CI [46.82, 68.70]) years.

The antimicrobial resistance of the *E. coli* strains was as follows: 100% were susceptible to carbapenems (i.e., meropenem, imipenem, and ertapenem) and nitrofurantoin; while 13 (43%) were ESBL producers and 17 (57%) were non-ESBL producers. Figure 1 depicts the distribution of ESBL-producing versus non-producing E. coli strains according to patient type (inpatient and outpatient) (*p* = 0.93). Table 1 compares the percentage of antimicrobial resistance of ESBL-producing versus non-ESBL-producing strains. There were no statistically significant differences between the percentages of resistance to ciprofloxacin (*p* = 0.376), trimethoprim-sulfamethoxazole (TMP-SMX) (*p* = 0.184), and gentamicin (*p* = 0.209) between ESBL-producing and non-ESBL-producing strains. Although comparison of resistances to nalidixic acid did not reach statistical significance (*p* = 0.083), resistance among non-ESBL-producing *E. coli* strains tended to be higher than among ESBL-producing *E. coli* strains.

Among the risk factors associated with infections by ESBL-producing *E. coli*, we observed that 100% of the strains registered at least one of the six risk factors assessed (Table 2). Regarding non-ESBL-producing strains, 35% did not register any of the risk factors analyzed (*p* = 0.021). The mean age of the patients with *E. coli* who did not carry ESBL was 49.23 years and of the ESBL carriers was 66.46 years (*p* = 0.023). The distribution of the strains by age group is observed in Table 2, identifying that the age group most affected by ESBL-producing strains was the one older than 80 years. No significant differences were found in the distribution by sex when comparing both groups.

We found that 92% (12/13) of the ESBL-producing strains carried the CTX-M type. CTX-M group 1 was found in 46% (6/13), (Table 3) whose variant CTX-M-15 was found in 83% (5/6) of them. In addition, CTX-M group 9 genes were found in 46% (6/13) and TEM was identified in 38% (5/13). No SHV genes were identified.

Table 3 describes the results from the molecular typing using Pasteur’s MLST technique. Of the 10 types of STs in the 30 *E. coli* samples analyzed, the most frequent were ST43, ST53, ST458 (17% each) and ST833 and ST3 (11% each). In 5.4% of the cases we identified ST4, ST479, ST526, ST621, and ST594. Among the ESBL-producing strains, ST43 of the Pasteur scheme—corresponding to ST131 in Achtman’s MLST scheme—was the most frequently identified ST (30%). The STs identified among outpatients were: ST43, ST53, ST458, and ST3.

## 4. Discussion

Worldwide, ESBL-producing *E. coli* strains have been increasingly reported in frequency and severity, making it an important ARO [2] that impacts the duration of hospital stays, delays appropriate antibiotic therapy, and increases healthcare costs [31]. In this study, *E. coli* strains isolated from clinical samples and showing a phenotype of antimicrobial resistance were analyzed. We observed that of the total isolates, 43% were producers of ESBL and 77% were resistant to ciprofloxacin, urinary tract infections being the main anatomical site of origin (67%) of the samples. In Latin America, the percentage of ESBL-producing *E. coli* strains is estimated at 24.7%, which has increased in the last decade, exceeded only by the Asia-Pacific region [24,31,32]. Regarding ciprofloxacin, the percentage of resistance in *E. coli* strains has been reported in 40.2% [31,32]. Panama reported for 2010 a percentage of ESBL-producing strains between 8–16% and 40.2% of strains resistant to fluoroquinolones among isolates of *E. coli* [24]. This high prevalence of resistance to these antimicrobials represents a great public health problem because both groups of antibiotics are widely used in the treatment of infections associated with *E. coli* [32].

An ST131 clone detected through the Achtman scheme accounts for nearly two-thirds of ESBL-producing isolates and for approximately 70 to 80% of fluoroquinolone-resistant isolates [33]. Our analysis using Pasteur’s MLST scheme showed the presence of this pandemic clone ST43/ST131 in 30% of the ESBL-producing *E. coli* strains, both in inpatients and outpatients. The ST43 clone has been described as specific for O25b/ST131, which was identified in 2008 as an important clone linked to CTX-M-15, strongly associated with fluoroquinolone resistance and co-resistance to fluoroquinolones, aminoglycosides, and TMP-SMX [34]. This coincides with our results that all the ST43 (ST131) strains identified in this study were resistant to ciprofloxacin (3/3), two strains (2/3) presented resistance to TMP/SMX and one strain (1/3) presented resistance to gentamicin. Within the three *E. coli* strains of the pandemic clone ST43/ST131 identified in this study, we found that one strain carried the CTX-M-15 enzyme and two strains carried enzymes corresponding to CTX-M-group 9. The literature describes that CTX-M15 (group-1) and CTX-M-14 (group-9) enzymes are the most commonly identified enzymes in the ST43/ST131 pandemic clone [16,17]. Recent Latin American studies have shown an increase in the prevalence of this group of enzymes in *E. coli* isolates [34,35,36,37]. In Latin America, clinical isolate studies have been carried out where the dissemination of the ST43/ST131 clone is evidenced in several countries both at the hospital and community level [17,23,24], however, in the Central American region there is little data regarding the identification of this clone. Studies have shown a prevalence of *E. coli* ST131 among non-ESBL-producing isolates of between 10 and 13% [8], however, in our analyzed strains, all strains belonging to this clone were carriers of ESBL. The frequency of ESBL-producing *E. coli* strains among inpatients and outpatients was quite similar (Figure 1). Previous studies have reported similar results indicating that ESBL-producing *E. coli* strains are increasing their dissemination in the community [4,38].

The migration of ESBL-producing Enterobacteriaceae from the hospital to the community setting is on the rise, for which the emergence and spread of CTX-M-type ESBLs throughout the world stands out [4]. Our data reflected that all patients from whom ESBL-bearing *E. coli* was isolated, were exposed to at least one risk factor associated with ESBL-producing *E. coli* infections in the community (Table 2), the most prevalent risk factors being a history of previous antibiotic use and previous hospitalization. Studies have shown that in adults with a history of previous antibiotic intake without specifying the classes, they present a risk factor for urinary infection due to ESBL-producing *E. coli* with an OR (odds ratio) that ranges between 3.1 and 5.6 in adults, while previous hospitalization has been evidenced as a risk factor with an OR that ranges between 1.7 and 3.9, which is influenced by the number of previous hospitalizations and the time elapsed [38]. Similarly, it has been described that people over 55 years of age without risk factors associated with medical care, have a risk two times higher (OR 2.05) of developing a UTI secondary to an ESBL-producing *E. coli* [39]. In our study, patients older than 80 years presented greater isolation of ESBL-producing *E. coli* strains.

Our analysis has shown the ST identified in the pediatric population as ST4, which showed resistance to ampicillin, ampicillin-sulbactam, and to TMP-SMX. Logan et al. [40] conducted a study in children and the identified ST of *E. coli* was ST4, which was described in the phylogroup B2. The authors described two ST4 isolates: in one the ESBL gene CTX-M-1 was identified and in the other, no ESBL genes were detected. This scenario alerts us to the potential difficulties in managing multidrug-resistant *E. coli* infections in pediatric patients.

As a limitation of the present study, we mention that it has been very difficult to obtain clinical samples due to the low number of requests for cultures by the treating physicians, as well as the low availability of human resources and infrastructure for the processing of cultures in the clinical laboratories of hospitals in Panama. Another limitation of this study is the small sample size, thus the interpretation for its implications, including the risk factor considerations should be prudent.

## 5. Conclusions

This study makes important contributions to the knowledge of the microbiology, molecular identification of β-lactamases, molecular epidemiology of *E. coli* strains in Panama, and identifies for the first time the pandemic clone ST43/ST131 harboring CTX-M-15 in the country, which calls our attention to the potential difficulties of treating infections of hospital and community origin, as well as the importance of knowing the composition and distribution of antibiotic resistance genotypes as an important step towards establishing public policies aimed at delimiting the impact of *E. coli* ARO infections in Panama.

## Figures and Tables

**Figure 1 antibiotics-10-00899-f001:**
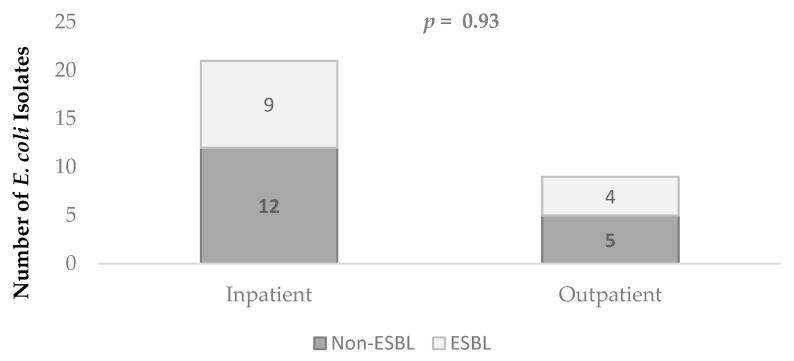
Distribution of ESBL-producing versus non-producing *E. coli* strains according to patient type. ESBL: extended-spectrum β-lactamase.

**Table 1 antibiotics-10-00899-t001:** Antimicrobial resistance of *Escherichia coli* isolates.

Antimicrobial Agent	MIC Breakpoint (µg/mL)	*Escherichia coli* Isolates by Resistance, *n* (%)		
		Total (*n* = 30)	ESBL (*n* = 13)	Non-ESBL (*n* = 17)
Ampicillin	≥32	27 (90)	13 (100)	14 (82)
Piperacillin-tazobactam	≥128/4	1 (3)	0 (0)	1 (6)
Cephalothin	≥64	20 (67)	13 (100)	7 (41)
Cefuroxime	≥64	16 (53)	13 (100)	3 (18)
Cefotaxime	≥64	13 (43)	13 (100)	0 (0)
Ceftazidime	≥64	13 (43)	13 (100)	0 (0)
Cefepime	≥64	13 (43)	13 (100)	0 (0)
Amikacin	≥16	2 (7)	1 (8)	1 (6)
Gentamicin	≥16	8 (27)	5 (38)	3 (18)
Nalidixic acid	≥32	17 (57)	5 (38)	12 (71)
Ciprofloxacin	≥4	23 (77)	11 (85)	12 (71)
Trimethoprim-sulfamethoxazole	≥320	19 (63)	10 (77)	9 (53)

ESBL: extended-spectrum β-lactamase; MIC: minimum inhibitory concentration.

**Table 2 antibiotics-10-00899-t002:** Identification of risk factors potentially associated with infections by ESBL-producing *Escherichia coli*.

Variables, *n* (%)	ESBL (*n* = 13)	Non-ESBL (*n* = 17)	*p* Value
**Age Groups, Years**			0.09
1–19	2 (15)	3 (18)	
20–59	1 (8)	3 (18)	
60–79	3 (23)	9 (53)	
≥80	7 (54)	2 (12)	
**Sex**			0.42
Female	9 (69)	10 (59)	
Male	4 (31)	7 (41)	
**Risk Factors**			
Hospitalized ≥2 d in the prior 90 d	8 (62)	7 (41)	0.28
Antibiotic treatment in the prior 90 d	7 (54)	5 (29)	0.50
Wound care at home	2 (15)	0 (0)	0.10
Outpatient chemotherapy	1 (8)	1 (6)	0.85
Personal history of immunosuppressive therapy	1 (8)	0 (0)	0.25
Hemodialysis in the prior 90 d	1 (8)	0 (0)	0.25
No known risk factors	0 (0)	6 (35)	0.021

d: days; ESBL: extended-spectrum β-lactamase.

**Table 3 antibiotics-10-00899-t003:** Phenotypic and genotypic characteristics of *Escherichia coli* isolates.

Isolate	Sequence Typing	ESBL	β-Lactamases	Originating Site	Source	Phenotypic Profile
378	3	+	CTX-M-group 9, TEM	H (Surg)	Wound	AMP, CEF, CFZ, CXM, CTX, CAZ, FEP, STX
2724	3	-		Outpatient	Urine	AMP, CEF
439	4	-		H (Peds)	Urine	AMP, STX
1232	43	+	CTX-M-group-1 (CTX-M-15)	H (Ort)	Wound	AMP, CEF, CFZ, CXM, CTX, CAZ, FEP, GEN, CIP
2690	43	+	CTX-M-group 9, TEM	Outpatient	Urine	AMP, CEF, CFZ, CXM, CTX, CAZ, FEP, NAL, CIP, STX
0-3630	43	+	CTX-M-group 9, TEM	Outpatient	Urine	AMP, CEF, CFZ, CXM, CTX, CAZ, FEP, NAL, CIP, STX
640	53	+	CTX-M group 9	H (Surg)	Urine	AMP, CEF, CFZ, CXM, CTX, CAZ, FEP, CIP
370	53	-		H (Ob-Gyn)	Urine	AMP, TZP, CEF, CXM, CAZ, AMK, NAL, CIP
2685	53	-		Outpatient	Urine	AMP, CEF, NAL, CIP, STX
2710	458	+	CTX-M-group 9	Outpatient	Urine	AMP, CEF, CFZ, CXM, CTX, CAZ, FEP, NAL, CIP, STX
19-2410	458	-		H (IM)	Blood	AMP, CIP, STX
2699	458	-		Outpatient	Urine	AMP, CEF, NAL, CIP
3627	479	-		H (Surg)	Urine	AMP, GEN, NAL, CIP, STX
382	526	+	CTX-M-group-1 (CTX-M-15)	H (IM)	Wound	AMP, CEF, CFZ, CXM, CTX, CAZ, FEP, STX
HRV-09	594	+	ND	I (ICU)	Wound	AMP, CEF, CFZ, CXM, CTX, CAZ, FEP, GEN, CIP, STX
361	621	+	CTX-M-group-1 (CTX-M-15)	H (IM)	Urine	AMP, CEF, CFZ, CXM, CTX, CAZ, FEP, CIP, STX
2676	833	+	CTX-M-group-1, TEM	Outpatient	Urine	AMP, CEF, CFZ, CXM, CTX, CAZ, FEP, GEN, NAL, CIP, STX
542	833	-		H (IM)	Blood	AMP, NAL, CIP, STX
375	N/A	+	CTX-M-group-1 (CTX-M-15), TEM	H (Surg)	Urine	AMP, CEF, CFZ, CXM, CTX, CAZ, FEP, NAL, GEN, CIP
638	N/A	+	CTX-M-group-1 (CTX-M-15)	H (Surg)	Urine	AMP, CEF, CFZ, CXM, CTX, CAZ, FEP, GEN, CIP, STX
CC2	N/A	+	CTX-M-group 9	H (IM)	Urine	AMP, CEF, CFZ, CXM, CTX, CAZ, FEP, CIP, STX
435, 655	N/A	-		H(IM)	Urine	AMP, NAL, CIP, STX
543, 544	N/A	-		H (Ob-Gyn)	Blood	AMP, NAL, CIP, STX
545	N/A	-		H (IM)	Blood	AMP, CIP, STX
O-2115	N/A	-		Outpatient	Urine	AMP, CIP, STX
CC1	N/A	-		H (IM)	Urine	NAL
519	N/A	-		H (Ob-Gyn)	Wound	AMP, NAL
436	N/A	-		Outpatient	Urine	AMP

AMP: ampicillin; AMK: amikacin; CAZ: ceftazidime; CEF: cephalothin; CFZ: cefazoline; CIP: ciprofloxacin; CTX: cefotaxime; CXM: cefuroxime; ESBL: extended-spectrum β-lactamase; FEP: cefepime; GEN: gentamicin; ICU: intensive care unit ward; IM: internal medicine ward; NAL: nalidixic acid; SXT: trimethoprim-sulfamethoxazole; TZP: piperacillin-tazobactam; ND: not detected; Ob-Gyn: obstetrics and gynecology ward; Ort: orthopedics ward; Peds: pediatric ward; Surg: surgery ward; +: ESBL; -: non-ESBL; N/A: not applicable.

## Data Availability

The data presented in this study are available within the article.

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
