# Peer review of "Molecular Epidemiology of Escherichia coli Clinical Isolates from Central Panama"

_antibiotics, 2021, doi:10.3390/antibiotics10080899_

Round 1
Reviewer 1 Report
The manuscript address one of the most common issues involved in the rise of MDRs. It is very crucial to conduct such surveillance studies to keep a track of the resistance bugs in a geographical setting which can be otherwise clinically challenging.
Minor comment:
If possible please re-write section 2.3, certain parts are repetitive .
Author Response
REVIEWER 1
Comments and Suggestions for Authors
The manuscript address one of the most common issues involved in the rise of MDRs. It is very crucial to conduct such surveillance studies to keep a track of the resistance bugs in a geographical setting which can be otherwise clinically challenging.
Minor comment:
If possible please re-write section 2.3, certain parts are repetitive.
Response: Thanks for your valuable comment. We have amended section 2.3 by removing the sentence that was repetitive.

Reviewer 2 Report
This manuscript describes the molecular epidemiology of clinical E. coli in Panama. The authors analyzed the diversity and prevalence of clinical E. coli isolates, putting emphasis on ESBL producers, from an undetermined number of isolates coming from two hospitals they found 30 that were resistant to at least one antibiotic, 13 of them encoded ESBL. The authors claim that in this study they characterize the clinical E. coli population in Panama (lines 79-81), however the presented data are from two hospitals and only include 30 E. coli isolates, I have the impression that the authors overinterpret the data they have. I think they should reformulate the purpose of the study. I regret to say that this manuscript is not suitable for publication at this stage.
Major comments:
In the introduction they present the ST43/ST131 clone, however they omit some central works about this clone like Price et al. 2013 (MBio), Petty et al. 2014. (PNAS), Stoesser et al. 2016 (MBio), and Ben Zakour et al. 2016 (MBio).
Why do the authors selected Pasteur’s MLST scheme over the Atchman which is most widely used? The authors mention that Pasteur ST43 is the pandemic clone ST131 (Atchman), It would be also interesting put the equivalences of the other Pasteur ST types mentioned in the manuscript to identify other disseminated ESBL associated clones like the ST167, ST617 for example.
In the Table 3 there are described only 21 isolates 9 are absent, why?.
The interpretation of the manuscript would be much easier if the authors prepare some figures, a minimum spanning three for example.
Minor comments:
Line 138-9: Which proportion represented these 30 isolates? how many samples were analyzed in total?
Line 152-153: Despite that there is a tendance I don’t think that the phrase “it is noteworthy that resistance was more frequent among non-ESBL-producing E. coli strains” is adequate.
Line 181: (Table 3) In the paper the authors mention 30 isolates, however in the table there are only 21.
Line 190-191: Not clear, please reformulate the phrase.
Line 192-193: “urinary tract infections being the main anatomical site of origin” revise English.
Line 201-203: Please cite some bibliography associated to this statement.
Line 251: Remove the reference to central America.
Author Response
REVIEWER 2
Comments and Suggestions for Authors
This manuscript describes the molecular epidemiology of clinical E. coli in Panama. The authors analyzed the diversity and prevalence of clinical E. coli isolates, putting emphasis on ESBL producers, from an undetermined number of isolates coming from two hospitals they found 30 that were resistant to at least one antibiotic, 13 of them encoded ESBL. The authors claim that in this study they characterize the clinical E. coli population in Panama (lines 79-81), however the presented data are from two hospitals and only include 30 E. coli isolates, I have the impression that the authors overinterpret the data they have. I think they should reformulate the purpose of the study. I regret to say that this manuscript is not suitable for publication at this stage.
Major comments:
In the introduction they present the ST43/ST131 clone, however they omit some central works about this clone like Price et al. 2013 (MBio), Petty et al. 2014. (PNAS), Stoesser et al. 2016 (MBio), and Ben Zakour et al. 2016 (MBio).
Response: Thanks for your valuable comment. We have now used in the introduction the references suggested by the reviewer to present the E. coli clone ST131 (Lines 63 to 89). We have included these articles in the references and amended the order of the references as needed.
Why do the authors selected Pasteur’s MLST scheme over the Atchman which is most widely used? The authors mention that Pasteur ST43 is the pandemic clone ST131 (Atchman), It would be also interesting put the equivalences of the other Pasteur ST types mentioned in the manuscript to identify other disseminated ESBL associated clones like the ST167, ST617 for example.
Response: Thanks for your comment and suggestion. We have used Pasteur's scheme since the strains belonging to Pasteur's ST43 correspond to serotype O25b of clone ST131 of E. coli, which is the most widespread and is related to greater virulence and pathogenicity. This point is important in this first study on the characterization of E. coli strains in Panama.
In the Table 3 there are described only 21 isolates 9 are absent, why?.
Response: We have now included in Table 3 the 30 isolates analyzed in this study. In 12 isolates there is no ST available (N/A) due to technical problems with the quality of the sequences, however, this issue does not change the interpretation of the results.
The interpretation of the manuscript would be much easier if the authors prepare some figures, a minimum spanning three for example.
Response: Thank you for this comment. A figure has been included on the distribution of BLEE-producing versus non-producing E. coli strains according to patient type (inpatient or outpatient). We have discussed the figure (lines 259-262) which now read: “The frequency of ESBL-producing E. coli strains among inpatients and outpatients was quite similar (Figure 1). Previous studies have reported similar results indicating that ESBL-producing E. coli strains are increasing their dissemination in the community.”
Minor comments:
Line 138-9: Which proportion represented these 30 isolates? how many samples were analyzed in total?
Response: Thank you for your question. We have rephrased this line hoping that now it is clear that there are 30 isolates analyzed in total (Line 133).
Line 152-153: Despite that there is a tendance I don’t think that the phrase “it is noteworthy that resistance was more frequent among non-ESBL-producing E. coli strains” is adequate.
Response: Thank you for your comment. We have rephrased this for clarity in the results (lines 205-208), which now read: “Although comparison of resistances to nalidixic acid did not reach statistical significance (P=0.083), resistance among non-ESBL–producing E. coli strains tended to be higher than among ESBL–producing E. coli strains.”
Line 181: (Table 3) In the paper the authors mention 30 isolates, however in the table there are only 21.
Response: Thanks for your valuable comment. We have now included in Table 3 the 30 isolates analyzed in this study. Please note that for 12 isolates there is no ST available (N/A) due to technical problems with the quality of the sequences, however, this issue does not change the interpretation of the results.
Line 190-191: Not clear, please reformulate the phrase.
Response: Thanks for your valuable comment. We have rephrased the sentence (line 226-228) which now read: “We observed that of the total isolates, 43% were producers of ESBL and 77% were resistant to ciprofloxacin, being urinary tract infections the main anatomical site of origin (67%) of the samples.”
Line 192-193: “urinary tract infections being the main anatomical site of origin” revise English.
Response: We have rephrased the sentence (line 226-228) which now read: “We observed that of the total isolates, 43% were producers of ESBL and 77% were resistant to ciprofloxacin, being urinary tract infections the main anatomical site of origin (67%) of the samples.”
Line 201-203: Please cite some bibliography associated to this statement.
Response: We have now cited a new article and rephrased the sentence to: “ST131 clone detected through the Achtman scheme accounts for nearly two-thirds of ESBL-producing isolates and for approximately 70% to 80% of fluoroquinolone-resistant isolates.”
Reference:
Banerjee, R, Johnson JR. A new clone sweeps clean: the enigmatic emergence of Escherichia coli sequence type 131. Antimicrob Agents Chemother. 2014, 58(9), 4997-5004. doi: 10.1128/AAC.02824-14
Line 251: Remove the reference to central America.
Response: We have removed the reference to central America

Reviewer 3 Report
This ms reports molecular epidemiological features of clinical ESBL-/no ESBL-producing E. coli isolates from Panama. Some risk factors were discussed. Despite widely available relevant studies in literature in the field, the findings are considered helpful for understanding the regional antibiotic resistance in E. coli. The approach taken are moderate, however, the reviewer acknowledges the resource limitations from the authors. A major concern is that the ms should more specify its limitations. For instance, in L245-247 (It is of importance to note the mentioning of the limitation), there is a need to more clearly point out the limitation of the number of isolates (with merely a total of 30 containing only 13 ESLBL producers). One major limitation is the limited number of the isolates, and thus the interpretation for the implications of the study including the risk factor consideration should be prudent.Additional changes (including editorial comments) are included for author’s consideration.
L22. Write Greek “β” for “Beta” and Change “lactam” to “Lactam” (to have “β-Lactam”).
L49. Change “the principal resistance mechanism” to “the principal mechanism of resistance to β-lactams”.
L114. Delete full spelling “Multilocus Sequence Typing” and Use “MLSI” as the abbreviation is already use in Line 64.
L115-116. Italicize all 8 gene names in the two lines.
L117. Introduce the commonly-use “PCR” and use it in L133 and other places in the ms to replace “polymerase chain reaction”.
L118-119. Delete “eight housekeeping” to avoid repetition.
L121-130. Delete all these descriptions as already in L113-121. Must be an error.
L134. Remove the comma before “were”.
L143. Change “sensitive” to “susceptible”.
Table 3: 21 isolates are present. It is unclear how these 21 are decided to be shown since there are a total of 30 isolates studied. Please clarify. A minor, for abbreviation of “AM” (ampicillin), please use three-letter abbreviation for consistency with other agents. Last column (Phenotypic Profile), formatting issue, do not all comma as the beginning of the sentence (row).
L220. Italicize “E. coli”.
L229-230. Per the PDF version, there is a formatting error. Please correct. What is “preto avoid repetition”? (totally unclear).
L284/366. Ref 3 and 34. Please use Greek “β” for “beta” as published in that cited journals.
L313. Ref 14 is incomplete with no authors or source. Please correct.
L339. Ref 24’s tilted. Please change “b” to Greek “β” (beta).
L344/354/360.Ref 26 and 30, 32’s titles. Please use lowercase sentence as other cited references for consistency. Also please use Greek “β” for “beta”.
Author Response
REVIEWER 3
Comments and Suggestions for Authors
This ms reports molecular epidemiological features of clinical ESBL-/no ESBL-producing E. coli isolates from Panama. Some risk factors were discussed. Despite widely available relevant studies in literature in the field, the findings are considered helpful for understanding the regional antibiotic resistance in E. coli. The approach taken are moderate, however, the reviewer acknowledges the resource limitations from the authors. A major concern is that the ms should more specify its limitations. For instance, in L245-247 (It is of importance to note the mentioning of the limitation), there is a need to more clearly point out the limitation of the number of isolates (with merely a total of 30 containing only 13 ESLBL producers). One major limitation is the limited number of the isolates, and thus the interpretation for the implications of the study including the risk factor consideration should be prudent. Additional changes (including editorial comments) are included for author’s consideration.
Response: Many thanks for your valuable comment. We did mention of the limitation related to the small sample size (lines 288-290) as suggested by the reviewer: “limitation of this study is the small sample size, thus the interpretation for its implications, including the risk factors consideration should be prudent.”
L22. Write Greek “β” for “Beta” and Change “lactam” to “Lactam” (to have “β-Lactam”).
Response: Thank you for this suggestion. This has now been changed along the revised manuscript from “β” to “Beta” and from “lactam” to “Lactam”.
L49. Change “the principal resistance mechanism” to “the principal mechanism of resistance to β-lactams”.
Response: Thank you for this suggestion. This has now been changed to “the principal mechanism of resistance to β-lactams”.
L114. Delete full spelling “Multilocus Sequence Typing” and Use “MLSI” as the abbreviation is already use in Line 64.
Response: This has now been changed to MLST (line 119-120).
L115-116. Italicize all 8 gene names in the two lines.
Response: We have italicized all 8 gene names in the two lines (lines 121-122).
L117. Introduce the commonly-use “PCR” and use it in L133 and other places in the ms to replace “polymerase chain reaction”.
Response: We have introduced the commonly-use “PCR” throughout the manuscript.
L118-119. Delete “eight housekeeping” to avoid repetition.
Response: We have deleted “eight housekeeping” to avoid repetition (line 124).
L121-130. Delete all these descriptions as already in L113-121. Must be an error.
Response: Thank you for highlighting this error. We have amended section 2.3 by removing the sentence that was repetitive.
L134. Remove the comma before “were”.
Response: We have removed the comma before “were” (line 130).
L143. Change “sensitive” to “susceptible”.
Response: We have now changed “sensitive” to “susceptible”.
Table 3: 21 isolates are present. It is unclear how these 21 are decided to be shown since there are a total of 30 isolates studied. Please clarify. A minor, for abbreviation of “AM” (ampicillin), please use three-letter abbreviation for consistency with other agents. Last column (Phenotypic Profile), formatting issue, do not all comma as the beginning of the sentence (row).
Response: Thanks for your valuable comment. We have now included in Table 3 the 30 isolates analyzed in this study. In 12 isolates there is no ST available (N/A) due to technical problems with the quality of the sequences, however, this issue does not change the interpretation of the results. We have used the three-letter abbreviation of “AMP” for consistency with other agents. We have amended the formatting issue in last column (Phenotypic Profile).
L220. Italicize “E. coli”.
Response: We have italicized E. coli throughout the manuscript.
L229-230. Per the PDF version, there is a formatting error. Please correct. What is “preto avoid repetition”? (totally unclear).
Response: We regret that we cannot find “preto avoid repetition”. We were not able to correct this error.
L284/366. Ref 3 and 34. Please use Greek “β” for “beta” as published in that cited journals.
Response: Thank you for pinpoint this matter. We have used greek “β” for “beta” throughout the manuscript and their references.
L313. Ref 14 is incomplete with no authors or source. Please correct.
Response: Reference has been completed. It is now reference number 18: Lahlaoui, H.; Ben Haj Khalifa, A.; Ben Moussa, M. Epidemiology of Enterobacteriaceae producing CTX-M type extended spectrum β-lactamase (ESBL). Med Mal Infect 2014, 44(9), 400-4. doi: 10.1016/j.medmal.2014.03.010
L339. Ref 24’s tilted. Please change “b” to Greek “β” (beta).
Response: We have used greek “β” throughout the manuscript and their references.
L344/354/360.Ref 26 and 30, 32’s titles. Please use lowercase sentence as other cited references for consistency. Also please use Greek “β” for “beta”.
Response: We have used lowercase sentence as other cited references for consistency and greek “β” throughout the manuscript and their references.

Round 2
Reviewer 2 Report
Dear editor
The authors have addressed all the questions I think the manuscript is ready now for publication.
sincerely